# Small Peptides Isolated from Enzymatic Hydrolyzate of *Pneumatophorus japonicus* Bone Promote Sleep by Regulating Circadian Rhythms

**DOI:** 10.3390/foods12030464

**Published:** 2023-01-19

**Authors:** Junbao Wang, Lu Zhang, Ningping Tao, Xichang Wang, Shanggui Deng, Mingyou Li, Yao Zu, Changhua Xu

**Affiliations:** 1College of Food Science & Technology, Shanghai Ocean University, Shanghai 201306, China; 2Shanghai Engineering Research Center of Aquatic-Product Processing & Preservation, Shanghai 201306, China; 3Laboratory of Quality and Safety Risk Assessment for Aquatic Products on Storage and Preservation, Ministry of Agriculture, Shanghai 201306, China; 4National R & D Branch Center for Freshwater Aquatic Products Processing Technology, Shanghai 201306, China; 5College of Food and Pharmacy, Zhejiang Ocean University, Zhoushan 316022, China; 6College of Fisheries and Life Science, Shanghai Ocean University, Shanghai 201306, China

**Keywords:** zebrafish, insomnia, sleep-promoting, peptides, transcriptomics, infrared transmission imaging

## Abstract

Due to the high addiction and side effects of medicines, people have increasingly inclined to natural and healthy peptides to improve sleep. Herein, we isolated novel peptides with sleep-promoting ability from *Pneumatophorus japonicus* bone peptides (PBPs) and constructed an insomniac zebrafish model as a demonstration, incorporating behavioral and transcriptomic approaches to reveal the sleep-promoting effect and mechanism of PBPs. Specifically, a sequential targeting isolation approach was developed to refine and identify a peptide with remarkable sleep-promoting activity, namely TG7 (Tyr-Gly-Asn-Pro-Trp-Glu-Lys). TG7 shows comparable effects and a similar action pathway to melatonin in improving sleep. TG7 restores abnormal behavior of insomnia zebrafish to normal levels by upregulating the hnrnpa3 gene. The peptide downregulates per1b gene but upregulates cry1b, cry1ba and per2, improving the circadian rhythm. Furthermore, TG7 upregulates the genes gnb3b, arr3b and opn1mw1 to regulate the visual function. The above results indicate that TG7 improves circadian rhythms and attenuated abnormal alterations in visual function and motility induced by light, allowing for effective sleep promotion. This study isolated sleep-promoting peptides from PBPs, which provides a theoretical basis for the development of subsequent sleep-promoting products based on protein peptides.

## 1. Introduction

When people are in a state of sleep, their brain and body get rest, recuperation and recovery, and the right amount of sleep is a prerequisite for healthy daily life. Sleep is essential for maintaining the balance of the internal body environment and regulating mood [1]. However, sleep quality is susceptible to various internal and external factors such as stress in life, negative emotions and changes in biological rhythms. Sleep deprivation can lead to a decrease in immunity [2], a lack of blood supply to the brain [3], memory loss [4] and in severe cases, it can lead to sleep disorders. Many diseases occur with varying degrees of sleep disturbance, such as cardiovascular disease [5], heart disease [6] and Parkinson’s disease [7]. Currently, the treatment of sleep disorders is mainly based on prescription medications such as benzothiazole receptor agonists and melatonin receptor agonists [8], supplemented by counseling therapy and physical therapy. However, due to the highly addictive nature of drugs and various side effects, a growing tendency to improve sleep is using natural and healthy active substances from nature, amongst which bioactive peptides derived from natural species have become one of the best choices.

Bioactive peptides have an important role in improving human health, and the potential physiological functions of bioactive peptides derived from proteins in traditional diets and medicinal herbs have attracted widespread scientific interest and attention. Many studies have shown that peptides have specific biological activities, such as antibacterial, antitumor and neuroprotective [9,10,11]. Among them, the effects of peptides on higher brain functions are of increasing interest. Many peptides derived from plants and animals have been shown to have sleep-promoting activity, such as walnut hydrolysate showing sleep-promoting effects in rats and improved memory deficits caused by sleep deprivation [12]. The tryptic digest of αS1 casein promoted sleep by reducing the number of sleep-wake cycles and increasing the protein expression of the β1 subtype of the GABAA receptor in the rat hypothalamus [13], and soy protein peptides promoted sleep by increasing melatonin and 5-HT levels in mice [14]. *Mauremys mutica* plastron enzymatic peptide ameliorated the disruption of the neurotransmitter system and promotes sleep in PCPA-induced insomnia mice [15]. These studies have demonstrated the promising ability of protein peptides to improve sleep. However, amino acid sequences of peptides with sleep-promoting activity have not yet been accurately identified and relevant pivotal sleep-promoting mechanisms are still unknown. Therefore, this study isolated and identified a sleep-promoting peptide from *Pneumatophorus japonicus* bone peptides with a well-defined amino acid sequence and investigated its sleep-promoting mechanism from a genetic perspective.

*Pneumatophorus japonicus* rich in proteins have important implications for the development and function of skeletal tissue, the central nervous system and the immune system [16]. *Pneumatophorus japonicus* also has the ability to treat neurological disorders and is rich in Docosahexaenoic acid (DHA). DHA is essential for the neurological development of the brain in infants and children, so it may be an ideal source of sleep-promoting peptides. In addition, we chose zebrafish as the experimental subject for this study. The genes in zebrafish are 87% homologous to human genes, and zebrafish also have a distinct circadian rhythm [17]. They are mainly diurnal species, but their rhythms are strongly linked to light, temperature and thermal cycles. It was found that fish brought into the light-dark cycle spend more than 75% of their daily activity during the light period, a diurnal pattern that occurs early in the life of zebrafish larvae [18]. Long-term light can disrupt the circadian balance in zebrafish, making them abnormally excited and active and triggering insomnia. Thus, the zebrafish has become an attractive model to study circadian and homeostatic processes that regulate sleep [19].

In this study, a targeted isolation approach was developed to prepare and refine sleep-promoting peptides (PBPs) from residues of *Pneumatophorus japonicus bones*. Combined with defined transcriptomics, young zebrafish with insomnia was selected as the research object and behavioral parameters were used as the activity evaluation indexes for multidimensional exploration of the sleep-promoting effect and mechanism of PBPs from full-image to molecular perspectives.

## 2. Materials and Methods

### 2.1. Materials and Reagents

*Pneumatophorus japonicus bones*: the residual waste of *Pneumatophorus japonicus* after meat picking and head removal, which was provided by the Zhejiang industrial group. TG7 (Tyr-Gly-Asn-Pro-Trp-Glu-Lys) and TG7-Acp-FITC were synthesized by GL Biochem (Shanghai) Ltd. Melatonin was purchased from Gaoxin chemical glass Co., Ltd. (Shanghai, China). Melatonin was dissolved in dimethyl sulfoxide (Shanghai, China) to make stock solutions.

### 2.2. Zebrafish Husbandry

The wild-type adult zebrafish AB line was obtained from the Key Laboratory of Exploration and Utilization of Aquatic Genetic Resources, Ministry of Education, Shanghai Ocean University (Shanghai, China). Zebrafish were handled according to procedures of the Institutional Animal Care and Use Committee (IACUC) of Shanghai Ocean University (Shanghai, China) and maintained according to standard protocols (http://zfin.org). This research was approved by the IACUC (IACUC 20171009) of SHOU. The procedures and research methodology were approved by the Shanghai Ocean University Experimentation Ethics Review Committee (SHOU-DW-2016-002). Wild-type adult zebrafish were raised in a light/dark environment (L/D) for 14/10 h, and the temperature of the circulating water system was 28.5 °C. The pH was maintained at 6.8–7.2. At 8: 00 p.m. the day before spawning, adult fish were placed in mating tank at a male-to-female ratio of 1: 1, male and female were separated by inserting a baffle. The following 8: 00 to remove the baffle, male and female fish began to chase, and an hour later collected embryos. Embryos were obtained by natural mating and cultured in a constant temperature incubator at 28.5 °C until four days after fertilization (4dpf). During period, water was exchanged every morning and evening to keep the water clean and remove dead eggs and debris in time.

### 2.3. Drug Treatment and Grouping

Contemporized zebrafish juveniles were collected and randomly arrayed by pipetting, zebrafish juveniles in 48 well plates containing culture medium. The following groups were created: (1) blank control group (BCG), normal growth of zebrafish without adding any drugs; (2) insomnia control group (ICG), zebrafish were continuously stimulated with 200 lux light for 24 h; (3) positive drug (melatonin) group (MTG), zebrafish immersed in 10^−3^ mol/L melatonin solution for 24 h after light-induced insomnia; (4) peptides groups (PBPs), zebrafish were soaked in different peptides solution for 24 h after light-induced insomnia. The concentrations of each component isolated from PBPs were as follows. The concentrations of PBPs1, PBPs2, and PBPs3 were all 0.25 mg/mL. The concentrations of PBPs1a, PBPs1b, and PBPs1c were all 0.5 mg/mL. The concentration of TG7 was 0.2 mg/mL. The concentration of each protein peptide component was determined to be safe and sleep-promoting for zebrafish in the pre-experiment.

### 2.4. Behavior Analysis

The behavioral test was performed during the same time frame. The 48-well plate was placed in Daniovision (Noldus, Wageningen, The Netherlands) without light. The behavior of zebrafish in each well was monitored by Daniovision with a resolution of 1024 × 768 pixels at 25 frames per second. Eight juveniles were assigned to examine effect of each concentration of the compound. The recorded video images were subjected to Ethovision XT11 (Noldus) to measure behavior of zebrafish in each well. Monitoring duration was 1 h, and the total moving distance, swimming speed, active time and resting time are measured.

### 2.5. Preparation Method of PBPs

The raw *Pneumatophorus japonicus* bones were minced and distilled water was added in the ratio of water: *Pneumatophorus japonicus* bones = 3.62 according to the optimal results of the previous single-factor experiments in the laboratory. The total amount of enzyme added to papain (enzyme activity 900 U/mg) and trypsin (enzyme activity 250 U/mg) was 5000 U/g. The amount of enzyme added to the two enzymes was calculated based on the weight of the *Pneumatophorus japonicus* bones weighed and the enzyme activity. The enzymatic digestion was carried out in a 120-rpm water bath at 50 °C and pH 7.0 for 4 h, enzyme inactivation at 95 °C for 15 min after the end of enzymatic digestion. After the enzyme solution was cooled to room temperature, the enzyme solution was centrifuged at 10,000 r/min for 15 min at 4 °C. The supernatant was collected in a 500 mL conical flask and decolorized by adding 1.5% activated carbon in a water bath at 60 °C for 30 min. After filtering out the activated carbon, the *Pneumatophorus japonicus* bone peptides (PBPs) was obtained.

### 2.6. Targeted Isolation and Identification of Sleep-Promoting Peptides from PBPs

PBPs was separated and collected through ultrafiltration membrane with a molecular weight (MW) of 10k Da, 3k Da and 1k Da, the components PBPs1 (MW < 1k Da), PBPs2 (1k Da < MW < 3k Da) and PBPs3 (3k Da < MW < 10k Da) were obtained. After vacuum freeze-drying, PBPs powder was obtained. The optimal molecular weight of sleep-promoting activity was obtained by behavioral indicators, then three components were separated by Sephadex gel column. The best sleep-promoting component was obtained by determination of the sleep-promoting activity. The composition of this component was analyzed by LC-MS/MS and its structure was determined (Appendix A), obtaining predominant sleep-promoting peptides. The sleep-promoting peptides were synthesized by solid-phase synthesis technology, and their activity was measured and transcribed. In addition, a fluorescent peptide was made by linking TG7 to the fluorescent molecule fluorescein isothiocyanate (FITC) using a solid-phase synthesis technique using Acp. FITC molecule contains isothiocyanic acid, which was more active compared with carboxyfluorescein and can bind to free amino group of peptide or protein under alkaline (pH 9.0–9.5) conditions to form FITC-peptide conjugate to make fluorescent peptide. Zebrafish were immersed in the appropriate concentration of fluorescent peptide solution according to the same method, and whether TG7 entered the zebrafish was observed by the yellow-green fluorescence emitted through FITC.

### 2.7. Zebrafish Infrared (IR) Spectra Data Acquisition

The treated zebrafish were embedded in a homemade mold (length × width × height of approximately 1 cm × 1 cm × 3 cm) with freezing embedding agent OCT and the zebrafish were positioned with forceps. Embedding was carried out at room temperature for 1 h, and freeze-setting was performed in a refrigerator at −80 °C for 3 h. Finalized samples were sliced with a thickness of 5 um using a frozen sectioning machine, adhered to ZnSe transmission carrier sheets, and dried in an oven at 40 °C for 12 h. ZnSe transmission carrier sheet was placed on the transmission imaging accessory carrier table of the Fourier Transform Microscopy Imaging System. After finding an appropriate area within the optical microscope view, the infrared transmission imaging image was acquired. Each pixel point in the infrared transmission imaging figure was an IR spectrum. In each infrared transmission imaging, five spectra were extracted from the head, abdomen, and tail of each zebrafish, for a total of 15 spectra, and the average spectrum was calculated for analysis. Spectrogram acquisition parameter: wave number range of 4000–750 cm^−1^, resolution of 4 cm^−1^, pixel size of 6.25 um and 16 scans per pixel point. Infrared transmission imaging images of zebrafish samples were analyzed by spectrum image software, and the images used for analysis were obtained after the removal of atmospheric noise, and baseline correction. Fourier deconvolution of the amide I in the spectrogram was performed using Peak Fit 4.12 software. The Gaussian function was used for the second-order derivative split peak fitting. The minimum residual R2 was obtained, and the relative percentage content of the corresponding protein secondary structure was calculated from each subpeak’s integrated area.

### 2.8. Transcriptomic Analysis

#### 2.8.1. RNA-seq Extraction

Zebrafish after insomnia were immersed in TG7 for 24 h and then treated with liquid nitrogen. RNA was extracted from zebrafish using a triazole kit (Thermo Fisher Scientific, Waltham, MA, USA), and RNA samples were subsequently subjected to strict quality control, mainly using an Agilent 2100 bioanalyzer: precise detection of RNA integrity. The qualified RNA samples were used for library construction and sequencing, and the constructed library was sequenced by Illumina Hi Seq^TM^, which was completed by the Novo Gene company (Beijing, China). To ensure quality and reliability of data analysis, the raw data needed to be filtered. This included removing reads with adapters, removing reads containing N (N indicated that base information could not be determined) and removing low-quality reads (reads with Q phred ≤ 20 bases accounted for more than 50% of entire read length). Additionally, Q20, Q30 and GC contents were calculated for the clean data (Appendix A).

#### 2.8.2. Statistics of Differentially Expressed Genes (DEGs)

After sequencing reads and removing spliced and low-quality sequences, high-quality sequences were aligned to zebrafish genome. Reference genome and gene model annotation files were downloaded directly from the genome website, the reference genome index was created using Hisat2 v2.0.5, and the paired reads were aligned to the reference genome using Hisat2 v2.0.5. The proportion of genomes that could be matched exceeded 88% (Appendix A) and differential genes between groups were analyzed after gene matching was completed. Input data for differential gene expression analysis was the read count data obtained from the gene expression level analysis, and DESeq2 R package software was used to analyze [20].

#### 2.8.3. Gene Ontology (GO) and Kyoto Encyclopedia of Genes and Genomes (KEGG) Enrichment Analysis of DEGs

Cluster profiler R package implemented GO enrichment analysis of DEGs, in which correcting gene length biases with corrected *p* value less than 0.05 were considered significantly enriched by DEGs. KEGG was a database resource for understanding high-level functions and utilities of biological systems. The degree of statistical enrichment of DEGs in the KEGG pathway could be tested by the cluster profiler R package.

### 2.9. Statistical Analysis

All experiments were conducted in triplicate (*n* = 3) and an ANOVA test (using SPSS 23.0 statistical software) was used to analyze data. Significant differences between the means of parameters were determined by using the Tukey test to analyze the difference. A value of *p* < 0.05 was considered to indicate statistical significance. Analysis of IR spectral data by Spectrum’s software PerkinElmer.

## 3. Results

### 3.1. Targeted Isolation of Sleep-Promoting Peptides from PBPs

We used the progressive isolation and purification of PBPs by the targeted isolation approach to obtain amino acid sequences with high sleep-promoting activity from them. The enzymatic digests of PBPs1, PBPs2 and PBPs3 were enriched in amino acids (Appendix A). The combined use of papain and trypsin resulted in the progressive release of these amino acids such as Gly, Tyr, Lys, Thr, Glu, Arg and Phe. They improved the sleep state by directly or indirectly regulating neurotransmitter receptor levels in vivo [21,22,23]. After analyzing the amino acid composition of PBPs1, PBPs2 and PBPs3, the optimal MW segment for sleep-promoting activity was determined by activity experiments. Insomnia triggered anxiety behaviors in zebrafish similar to those in humans, as up to 87% of zebrafish genes were homologous to humans [24]. The anxious behavior of zebrafish was concretely reflected in behavioral indicators, including moving distance, swimming speed, active time and resting time, in which insomniac zebrafish showed perceptible differences from normal zebrafish. Compared with ICG, PBPs1, PBPs2 and PBPs3 could restore abnormal changes of behavioral indexes by insomnia in zebrafish, specifically, they shortened the moving distance by the zebrafish, reduced swimming speed, and active time, and increased resting time (Figure 1). PBPs1 had a significant effect on reducing moving distance, swimming speed and active time, and increasing resting time of zebrafish (*p* < 0.05). The MTG had the best effect, followed by PBPs1 > PBPs2~PBPs3, which indicated that low molecular weight peptides were more readily absorbed and utilized by zebrafish [25]. On this basis, the PBPs1 portion was further refined.

### 3.2. Analysis of IR Spectra of Zebrafish Treated with PBPs1

Infrared transmission imaging not only contained information on the spatial distribution of the sample in micro-regions, but also obtained information on the chemical composition of the sample, such as lipids, proteins and nucleic acids [26]. Behavioral experiments demonstrated the best sleep-promoting activity of PBPs1 to further investigate whether PBPs1 affected various components of lipids, proteins, etc., in insomniac zebrafish. The IR spectra of zebrafish after diverse treatments showed significant differences in peak intensities and shapes (Figure 2A), and two bands majorly attributed to lipids (3000–2800 cm^−1^) and proteins (1750–1450 cm^−1^) were selected for spectral analysis. 3000–2800 cm^−1^ reflected the information of lipid composition in zebrafish, which was formed by 2873 cm^−1^ (antisymmetric stretching motion of CH_3_) and 2925 cm^−1^ (symmetric stretching motion of CH_2_) and the vibration of some amino acid side chains. Compared to BCG, zebrafish in the ICG had the highest peak in lipid composition (Figure 2B), in agreement with the finding that sleep deprivation caused an increase in body fat content in individuals, resulting in central obesity [27]. PBPs1 had the same treatment effect as the sleep-promoting drug melatonin, as they both gradually decreased the peak in insomniac zebrafish at 2925 cm^−1^ and 2873 cm^−1^ to approach normal levels. The peak intensities of amide I (~1651 cm^−1^, vibrational stretching of C=O) and amide II (~1543 cm^−1^, bending vibration of N-H) in insomniac zebrafish were lower than those in the BCG (Figure 2C), and the peak intensities of amide I and amide II bands in insomniac zebrafish gradually increased and approached normal levels after PBPs1 treatment, but this effect was not observed in MTG. To further reveal the variation pattern of protein content in zebrafish under different treatment conditions, the SD-IR spectra at 1700–1450 cm^−1^ were extracted and analyzed (Figure 2D). The variation of peak intensities in SD-IR spectra can accurately represent content variance of components. Therefore, SD-IR spectra could be applied to further amplify the tiny differences of IR spectra. Compared with BCG, the α-helix (1658–1650 cm^−1^), random coil (1650–1640 cm^−1^), and β-sheet (1640–1610 cm^−1^) contents increased in insomniac zebrafish; the β-turn (1695–1660 cm^−1^) content decreased. The contents of α-helix, random coil, and β-sheet decreased after PBPs1 treatment; the content of β-turn increased and approached normal zebrafish (Figure 2E). These results revealed that PBPs1 ameliorated lipids and proteins adverse changes in insomniac zebrafish, and subsequently further isolated and purified PBPs1.

### 3.3. Evaluation of the Sleep-Promoting Activity of PBPs1

PBPs1 was separated into three fractions: PBPs1a, PBPs1b and PBPs1c, by chromatographic separation techniques (Appendix A). The isolated fractions were collected and lyophilized for sleep-promoting activity assays. Compared with ICG, PBPs1a, PBPs1b and PBPs1c treatments all restored physiological deviation induced by insomnia in zebrafish. PBPs1c and PBPs1a had a significant effect on reducing moving distance and slowing down swimming speed of insomniac zebrafish (*p* < 0.05); moving distance and swimming speed of each group were ranked as PBPs1c < PBPs1a < PBPs1b (Figure 3). The active time of zebrafish after PBPs1a, PBPs1b and PBPs1c treatment was lower than ICG and BCG, but the resting time was correspondingly higher than the two groups. These results showed that the peptides groups reduced active time and increased resting time as the MTG. Specifically, PBPs1c had the best sleep-promoting effect and thus was further identified and analyzed to screen for pure amino acid sequences with the optimal sleep-promoting effects. 

### 3.4. Sequence Identification of Sleep-Promoting Peptides in PBPs1c

The amino acid sequence and structure identification of PBPs1c component were carried out by LC-MS/MS. More than 20 amino acid sequences and their corresponding information were obtained by Peaks studio 10.0 software analysis (Appendix A). A higher value of the area in the sequence information from software analysis indicated a higher percentage of this amino acid sequence in the sample. The amino acid content was used as the main basis for screening, and the molecular weight deviation of this sequence was within 10 ppm. Screening based on area values, the best peptide sequence as Tyr-Gly-Asn-Pro-Trp-Glu-Lys (TG7) was identified. TG7 and TG7-Acp-FITC were synthesized by solid-phase synthesis technique, and the purity of TG7 was 98% by RP-HPLC and mass spectrometry, so they could be used for the determination of sleep-promoting activity.

### 3.5. Evaluation of the Sleep-Promoting Activity of TG7

The sleep-promoting activity of TG7 was characterized by behavioral experiments, thus verifying the accuracy of the LC-MS/MS identification results. TG7 improved the changes in abnormal behavioral indicators caused by insomnia in zebrafish compared to ICG. Specifically, this was evidenced by a significant reduction in moving distance by insomniac zebrafish, a decrease in swimming speed and active time and an increase in resting time (Figure 4A). From the behavioral trajectory plots of zebrafish in different treatment groups (Figure 4B), TG7 attenuated motility of insomniac zebrafish, resulting in a reduction in motility trajectory of zebrafish. Data from four behavioral metrics (moving distance, swimming speed, active time and resting time) were used to create a behavioral fingerprint of zebrafish (Figure 4C); the behavioral fingerprinting could be a good way to compare the variability or correlation between different treatment groups. Combining an analysis of behavioral fingerprinting and behavioral data, TG7 and ICG showed greater differences in various behavioral indicators. This demonstrated that TG7 had a better sleep-promoting effect on insomniac zebrafish and that the sleep-promoting effect was close to that of the MTG. Although the optimal concentration of TG7-treated zebrafish was 0.2 mg/mL, the appropriate concentration of TG7-Acp-FITC-treated zebrafish remained unclear. Therefore, a gradient concentration of TG7-Acp-FITC was set and its distribution in zebrafish was observed. TG7-Acp-FITC (0.05–0.15 mg/mL) could be absorbed into the abdomen of zebrafish to emit yellow-green fluorescence; when concentration started from 0.2 mg/mL, the head and eyes of zebrafish also emitted a small amount of fluorescence (Figure 4D). This indicated that TG7-Acp-FITC was absorbed into the body by zebrafish and transported to various parts of body through its digestion and catabolism. These results further revealed that TG7, when absorbed into zebrafish, improved sleep by restoring the abnormal behavioral ability of insomniac zebrafish and by regulating lipids and proteins content in the body. In a subsequent experiment, transcriptome analysis was performed on TG7-treated zebrafish to reveal the sleep-promoting mechanism of TG7.

### 3.6. Statistics of DEGs

The screening parameters of DEGs were set as |log2 fold change| > 0 and *p*-value < 0.05, with the change of gene expression ploidy in different samples as the marker; the volcano map was drawn in order of statistical significance of DEGs to visualize the DEGs and show their different expression patterns. In MTG, 1993 genes were differentially expressed, of which 941 genes were upregulated and 1052 genes were downregulated. In TG7, 1577 genes were differentially expressed, of which 786 were upregulated and 791 were downregulated (Figure 5A,B). A Venn diagram showed the overlap of DEGs among different comparative combinations, which could be used to screen for DEGs that were common or unique to certain comparative combinations. From the Venn diagram, it was found that there were 13014 shared DEGs and 692 exclusive DEGs in MTG and ICG, and 13074 shared DEGs and 640 exclusive DEGs in TG7 and ICG (Figure 5C,D). The Venn diagram results demonstrate that TG7 and MTG regulate similar numbers of DEGs in insomniac zebrafish that may contain sleep-related genes.

### 3.7. Screening of DEGs

Treatment with TG7 and MTG resulted in alterations in various genes in insomniac zebrafish. The similarities and differences of these genes were compared to explore the reasons for the ability of TG7 to promote sleep. The default was to use padj < 0.05, |log2 Fold Change| > 1 as difference significance criteria, 15 DEGs with greater significance were selected from MTG and TG7, respectively, for comparison (Table 1). Compared with ICG, both MTG and TG7 were able to upregulate the genes per2, cry1bb and cry1ba; but downregulate the gene per1b, which were part of the biological clock genes that regulated circadian rhythms in zebrafish [28]. Both MTG and TG7 upregulated the genes arr3b, opn1mw1, grk7a and gnb3b, which were associated with visual development in zebrafish [29], thus the two groups ameliorated circadian rhythm disturbance and eyes development retard of zebrafish resulting from insomnia triggered by light. Furthermore, TG7 could also upregulate the genes gabra6a, gngt2b (genes related to the metabolism of sleep neurotransmitter gamma-aminobutyric acid [30]) and hnrnpa3 (gene played important roles in regulating neural progenitor cell division and cerebral cortex development [31]), while downregulating the gene sik1 (regulation of circadian rhythms [32]).

### 3.8. GO and KEGG Enrichment Analysis of DEGs

After DEGs were obtained based on gene expression analysis, they must be further reduced to function of the genes. GO pathway enrichment was performed on DEGs produced after treatment of insomniac zebrafish in MTG and TG7, and the 30 most significant terms were selected from the GO enrichment results and bar graphs were drawn for display. Various groups of DEGs were enriched in three functional categories: cell composition (CC), biological processes (BP), and molecular functions (MF). Compared with ICG, BP was significantly enriched in the organonitrogen compound catabolic process; CC was significantly enriched in proteasome core complex; MF was significantly enriched in threonine-type endopeptidase activity after MTG treatment (Figure 6A). In TG7, compared with ICG, the significantly affected BP after the peptide treatment was mainly protein catabolic process; the significantly affected CC was proteasome core complex; the significantly affected MF was threonine-type endopeptidase activity (Figure 6B). The results of GO enrichment showed that DEGs could be enriched in proteasome core complex, threonine-type endopeptidase activity in zebrafish after treatment in both MTG and TG7.

Based on the screened DEGs, pathway enrichment analysis was performed with KEGG. The 20 most important terms from the KEGG enrichment results were selected to draw scatter plots for analysis. The results of DEGs generation from MTG and TG7 compared with ICG, and then KEGG pathway maps were generated. Compared to ICG, DEGs in MTG were significantly enriched in progesterone-mediated oocyte maturation; glycerolipid metabolism; MAPK signaling pathway (Figure 6C). The TG7 treatment of insomniac zebrafish resulted in a significant enrichment of certain DEGs in focal adhesion, alanine; aspartate and glutamate metabolism; arachidonic acid metabolism (Figure 6D). The signaling pathways that were significantly enriched by DEGs in both MTG and TG7 were phototransduction and proteasome, as evidenced by the KEGG enrichment results.

## 4. Discussion

This study developed a new approach for the targeted isolation of sleep-promoting peptides, combining bioactive analysis with chemical analysis. These mainly include: (1) *Pneumatophorus japonicus* bones were digested enzymatically by papain and trypsin (1:1); the enzymatic peptides (PBPs) were divided into different molecular weights by ultrafiltration membranes of 10K Da, 3K Da and 1K Da, and the target molecular weight of PBPs was precisely isolated by Sephadex G-15 gel; the simultaneous in vivo verification of sleep-promoting activity in combination with insomniac zebrafish models. (2) Food multimolecular spectroscopy was used to analyze zebrafish’s lipids and proteins composition changes [33]. (3) The sequences of the sleep-promoting amino acids in PBPs were identified by LC-MS/MS. The approach allowed a full multi-dimensional systematic study of aquatic processing by-products from raw material to final extract to the target. It had the advantages of a high targeted extraction rate and a high activity of target protein peptides. Besides, the approach allowed both overall composition analysis by IR projection imaging and IR spectroscopy, and chemical characterization and chemical fingerprinting by LC-MS/MS. Therefore, it was a targeted protein-peptide isolation approach that integrated chemical fingerprinting, multi-molecular infrared spectroscopy and bioactivity evaluation.

The amino acid sequence Tyr-Gly-Asn-Pro-Trp-Glu-Lys (TG7) with high sleep-promoting activity was identified from PBPs by this approach. TG7 linked with a fluorophore FITC was able to enter the yolk sac, head and eye sites of zebrafish and emit yellow-green fluorescence. Furthermore, combining behavioral and multi-molecular spectroscopic analysis, not only was the sleep-promoting activity of TG7 evaluated visually, but the overall composition analysis of TG7-treated insomniac zebrafish could also be performed. It was speculated that TG7 improved sleep by restoring the levels of lipids and proteins, and abnormal behavioral abilities in the body of insomniac zebrafish. Transcriptomic results showed that the positive drug group (MTG) and the peptide group (TG7) could regulate the same genes (per1b, per2, cry1bb, cry1ba, arr3b, opn1mw1, grk7a and gnb3b) in insomniac zebrafish, while differentially expressed genes (DEGs) could also be enriched in the same signaling pathways. Furthermore, TG7 also individually regulated genes (sik1, hnrnpa3, gabra6a and gngt2b) in insomniac zebrafish. Therefore, by analyzing the link between these genes and the related signaling pathways with sleep, the mechanism of sleep-promoting by TG7 was revealed (Figure 7).

By comparing the DEGs of TG7 and ICG, it can be shown that, like ICG, TG7 downregulates per1b, while upregulating cry1bb, cry1ba and per2, which were biological clock genes associated with circadian rhythms in zebrafish [34]. Zebrafish disrupted their own circadian rhythm after prolonged light treatment, causing insomnia and metabolic disorders in the body, which led to the low expression of cry1bb, cry1ba and per2 genes in zebrafish. Treatment with TG7 upregulated these genes, thereby restoring circadian rhythms to normal levels in insomniac zebrafish. It has been shown that zebrafish with the per1b mutant exhibit hyperactivity, decreased attention span and reduced dopamine secretion [35]. In this experiment, insomnia caused a high expression of per1b gene in zebrafish, which resulted in higher moving distance, swimming speed and activity time than normal zebrafish, and these behavioral indicators could be reduced and approached to normal levels after treatment with TG7. Therefore, TG7 could down-regulate per1b and thus improve the abnormal behavioral ability of insomnia zebrafish. In combination with GO enrichment pathway analysis, TG7 was significantly enriched in proteasome core complex, which was essential for the regulation of circadian rhythm and sleep homeostasis, compared to ICG. A recent study demonstrated that knocking out the Rpt2 gene in the Drosophila central nervous system reduced proteasome activity, increased the number of insoluble proteins, and led to a Parkinson’s-like symptom phenotype in Drosophila, manifesting as reduced viability, hyperactivity and sleep deprivation [36]. The production and accumulation of biological clock proteins were regulated by biological clock genes, and the proteasome-mediated degradation of core biological clock proteins and synaptic proteins contributed to the regulation of sleep quality [37], while an abnormal expression of biological clock genes caused by insomnia could lead to an excessive production and accumulation of biological clock proteins, the process that required a degradation of the proteasome.

Gnb3b, arr3b and opn1mw1 were expressed in dorsal and right sides of zebrafish retina to protect the lower part of retina from intense light damage [38], while grk7a was associated with light sensitivity in zebrafish eyes; all of these genes were associated with visual function in zebrafish. Prolonged light-induced visual cycle dysregulation affected circadian rhythm and the normal developmental function of the eye in zebrafish [39]. TG7 restored visual function in insomniac zebrafish by significantly upregulating the genes gnb3b, arr3b, grk7a and opn1mw1; KEGG enrichment results indicated that the signaling pathway in which these genes could be significantly enriched was phototransduction. The link between the mammalian circadian rhythms and phototransduction was primarily associated with melanopsin [40]. Circadian rhythms rhythmically regulated organisms’ physiology, behavior and sleep, which were closely related to the regular environment and time of day, depending on the natural cycles of light and temperature [41]. The insomnia model exposed to light for 24 h disrupted the natural cycle of light perception in zebrafish larvae. The expression of genes related to visual function in zebrafish larvae was affected, which also affected the phototransduction process. This led to circadian rhythm disturbance in zebrafish larvae triggering insomnia, and treatment with TG7 improved the sleep of insomniac zebrafish.

Compared with MTG, TG7 could also individually regulate certain genes such as sik1, hnrnpa3, gabra6a and gngt2b, which also played a role in the regulation of sleep. Sik1 had been reported to regulate circadian behavior and energy metabolism, tapping sik1 in the suprachiasmatic control nucleus led to a rapid phase shift in circadian rhythms; in other words, the sik1 gene was resistant to jet lag [42]. One of the reasons that the sik1 gene might increase the need for sleep was the removal of the protein kinase A site from sik1 [43]. Therefore, it could be speculated that TG7 caused a low expression of the sik1 gene, thereby increasing the sleep requirement of zebrafish. TG7 significantly upregulated hnrnpa3 in insomniac zebrafish. The hnrnpa3 gene played a crucial role in regulating the division of neural progenitor cells and the development of the cerebral cortex, which controlled various sensory and perceptual information inputs, motor control, learning and memory, and metabolism of the body in mammals. The moving distance and swimming speed of insomniac zebrafish could be restored to normal levels after treatment with the peptide group, which might also be regulated by hnrnpa3. GABA played an essential role as an inhibitory neurotransmitter in improving sleep quality, and its anabolism was regulated in many ways [44]. TG7 probably increased GABA content in zebrafish brain through upregulation of gabra6a and gngt2b genes, thereby improving sleep.

Overall, the sleep-promoting amino acid sequence Tyr-Gly-Asn-Pro-Trp-Glu-Lys (TG7) was progressively isolated and identified from *Pneumatophorus japonicus* bone peptides (PBPs) in this study, transcriptomics further revealed that TG7 promoted sleep by up-regulating or down-regulating genes related to circadian rhythm, visual function, and motility in insomniac zebrafish. This study not only increased the utilization rate of *Pneumatophorus japonicus* by-products, achieving the purpose of energy saving and green environment protection, but also identified new sleep-promoting peptides from PBPs, providing new thinking and theoretical basis for the clinical development of safe and effective protein peptide sleep-promoting products. The results of the transcriptomic part help us to understand the sleep-promoting mechanism of TG7 from a genetic perspective, but it is still necessary to reveal the sleep-promoting mechanism of TG7 from multiple perspectives and in a comprehensive manner. For example, changes in the content of sleep-related neurotransmitters in zebrafish after TG7 treatment can be examined in subsequent studies, as well as further evaluation of the drug safety of TG7.

## 5. Conclusions

In summary, it was demonstrated that the peptide TG7, identified from PBPs, had excellent effects on improving sleep in zebrafish. Light-induced insomnia adversely affected circadian rhythm, visual function and motility in zebrafish, and TG7 improved the abnormal expression of genes associated with circadian rhythm, visual function and motility in zebrafish. This study elucidated the sleep-promoting effect of TG7 and its potential mechanism, which provided a basis for the high-value utilization of *Pneumatophorus japonicus* and laid a good experimental foundation for exploring functional and medicinal food peptides with sleep-promoting ability.

## Figures and Tables

**Figure 1 foods-12-00464-f001:**
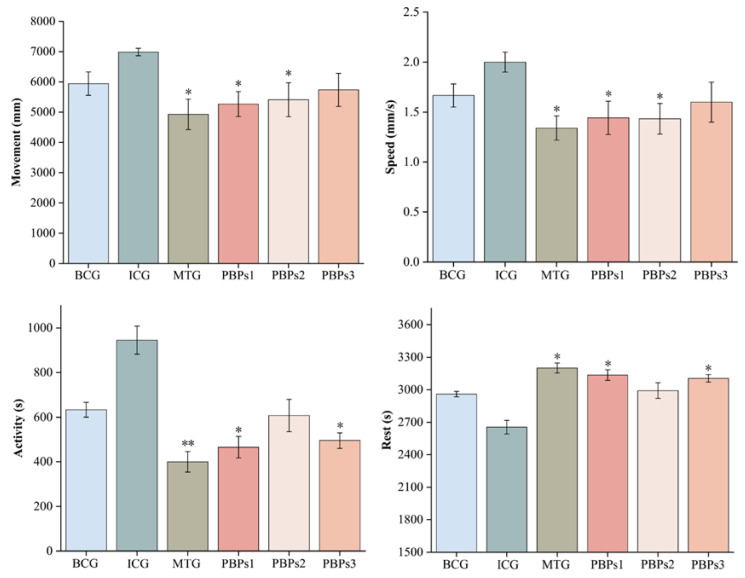
Changes in behavioral indicators of zebrafish after PBPs1 (0.25 mg/mL), PBPs2 (0.25 mg/mL) and PBPs3 (0.25 mg/mL) treatment. The *p*-values for each group were calculated by comparison with the ICG. Blank control group (BCG), normal growth of zebrafish without adding any drugs; Insomnia control group (ICG); Positive drug (melatonin) group (MTG). Data are expressed as mean ± SD. (Student *t*-test, * *p* < 0.05, ** *p* < 0.01, *n* = 8).

**Figure 2 foods-12-00464-f002:**
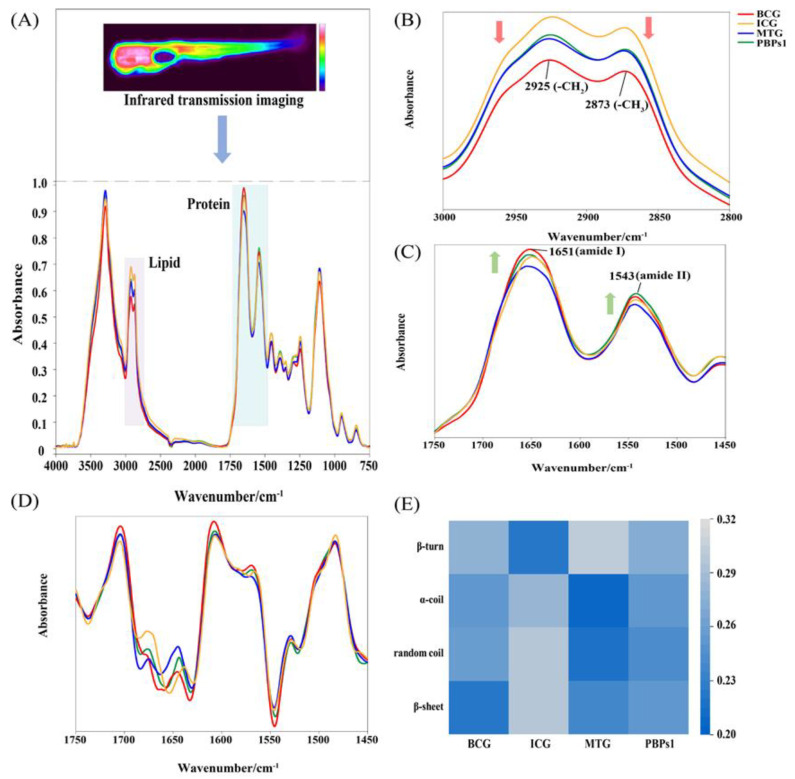
IR spectra of zebrafish at 4000–750 cm^−1^ (**A**); IR spectra of zebrafish at 3000–2800 cm^−11^ (**B**); IR spectra of zebrafish at 1750–1450 cm^−11^ (**C**); SD-IR spectra of zebrafish at 1750–1450 cm^−11^ (**D**); heat map of protein secondary structure occupancy (**E**). Blank control group (BCG), normal growth of zebrafish without adding any drugs; Insomnia control group (ICG); Positive drug (melatonin) group (MTG).

**Figure 3 foods-12-00464-f003:**
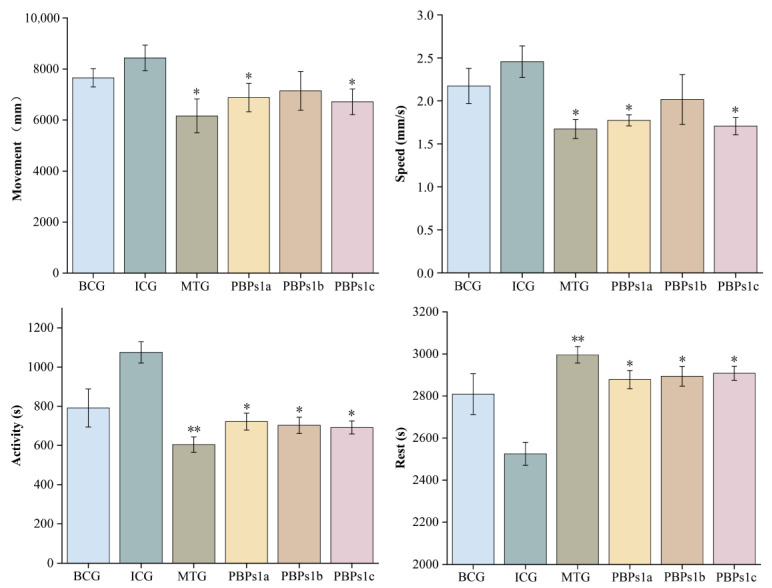
Changes in behavioral indicators of zebrafish after PBPs1a (0.5 mg/mL), PBPs1b (0.5 mg/mL) and PBPs1c (0.5 mg/mL) treatment. The *p*-values for each group were calculated by comparison with the ICG. Blank control group (BCG), normal growth of zebrafish without adding any drugs; Insomnia control group (ICG); Positive drug (melatonin) group (MTG). Data are expressed as mean ± SD. (Student *t*-test, * *p* < 0.05, ** *p* < 0.01, *n* = 8).

**Figure 4 foods-12-00464-f004:**
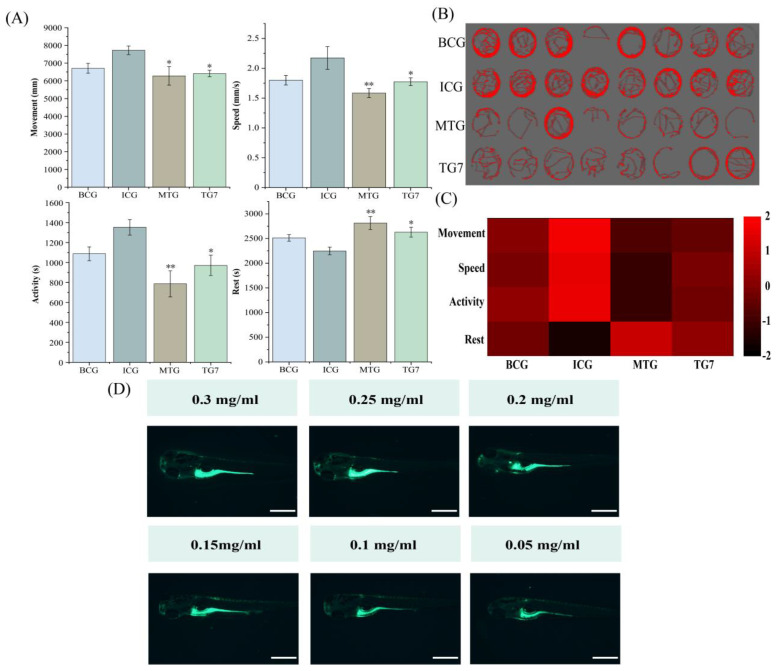
(**A**) Effect of TG7 (0.2 mg/mL) on behavioral indicators of zebrafish; (**B**) zebrafish behavior trajectory map; (**C**) behavioral fingerprinting; (**D**) distribution and fluorescence intensity of TG7-Acp-FITC in zebrafish. Scale bars: 500 µm. The *p*-values for each group were calculated by comparison with the ICG. Blank control group (BCG), normal growth of zebrafish without adding any drugs; Insomnia control group (ICG); Positive drug (melatonin) group (MTG); Tyr-Gly-Asn-Pro-Trp-Glu-Lys (TG7). Data are expressed as mean ± SD. (Student *t*-test, * *p* < 0.05, ** *p* < 0.01, *n* = 8).

**Figure 5 foods-12-00464-f005:**
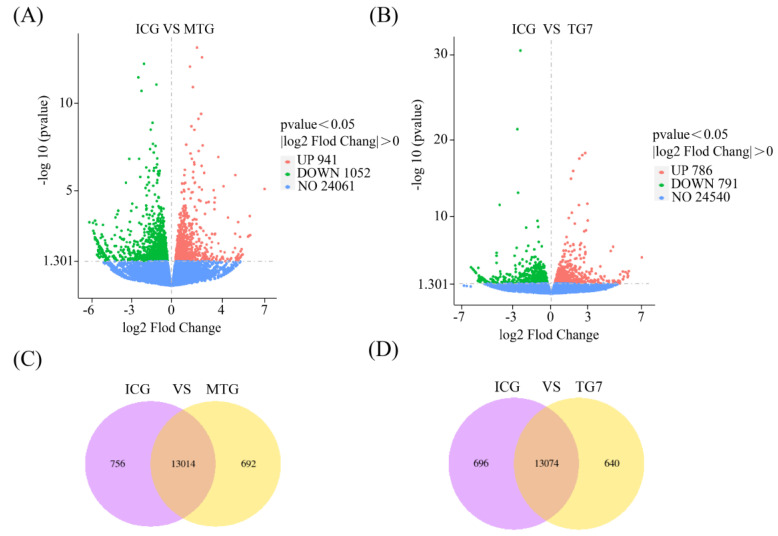
(**A**,**B**) Plots of DEGs volcanoes in MTG and TG7 compared with ICG. The horizontal coordinates indicated fold change of gene expression in the treated and control groups (log2 Fold Change), while the vertical coordinates indicated the significance level of DEGs in the treated and control groups (−log10 *p* value). Genes with obvious upward calls were indicated by red dots, genes with obvious downward calls were indicated by green dots, and genes without obvious differential expression were indicated by blue dots. (**C**,**D**) Venn diagram of DEGs in MTG and TG7 compared with ICG. The sum of all numbers in circles represents total number of DEGs in this comparison combination, overlapping areas indicated the number of DEGs shared between combinations. Insomnia control group (ICG); Positive drug (melatonin) group (MTG); Tyr-Gly-Asn-Pro-Trp-Glu-Lys (TG7).

**Figure 6 foods-12-00464-f006:**
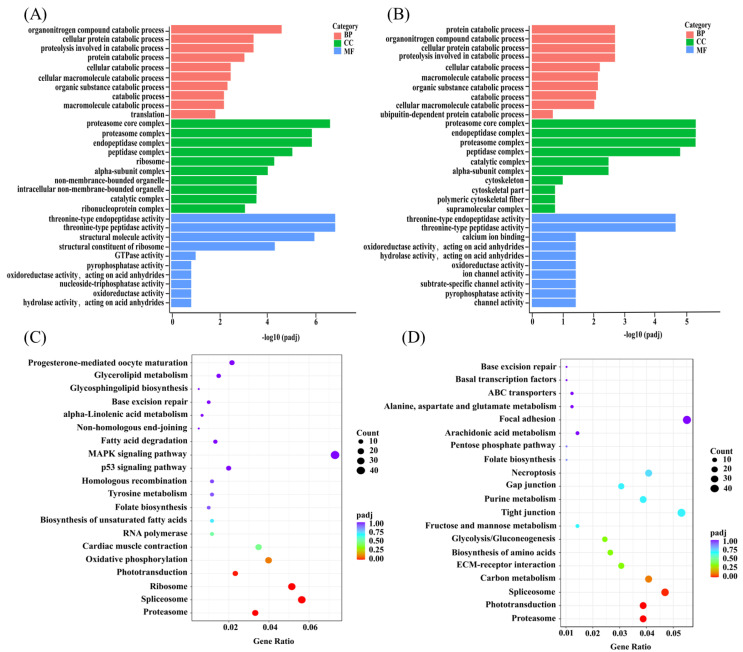
GO enrichment pathway of MTG (**A**) and TG7 (**B**). The horizontal coordinate was GO Term, the vertical coordinate is significance level of GO term enrichment, the higher value the more significant. KEGG enrichment pathway of MTG (**C**) and TG7 (**D**). The horizontal coordinate was ratio of the number of DEGs annotated to total number of DEGs on the KEGG pathway, vertical coordinate was the KEGG pathway, size of dot represents the number of genes annotated to the KEGG pathway, with the color ranging from red to purple representing significance of the enrichment. Insomnia control group (ICG); Positive drug (melatonin) group (MTG); Tyr-Gly-Asn-Pro-Trp-Glu-Lys (TG7).

**Figure 7 foods-12-00464-f007:**
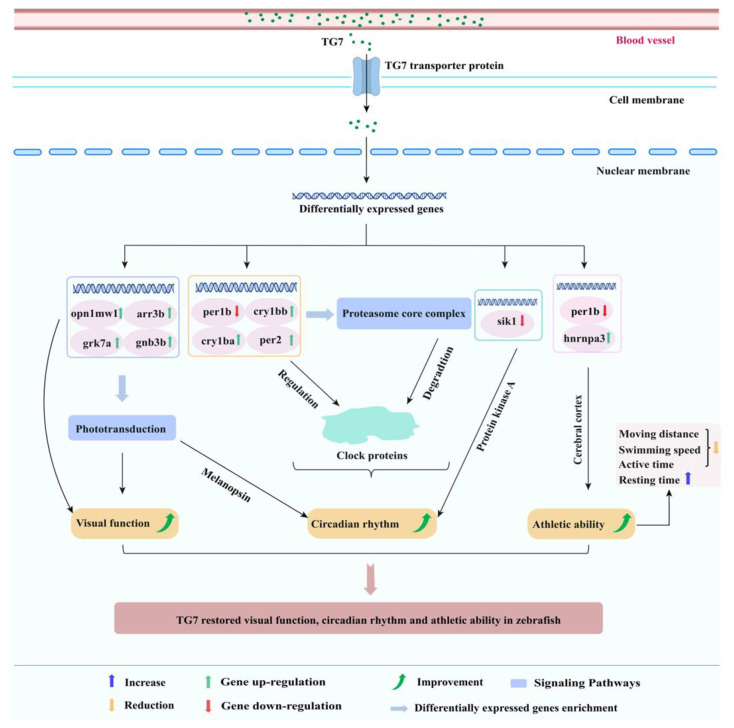
Sleep-promoting mechanism of TG7 on zebrafish. Tyr-Gly-Asn-Pro-Trp-Glu-Lys (TG7).

**Table 1 foods-12-00464-t001:** Expression result comparison of 15 genes between MTG and TG7 with ICG. Insomnia control group (ICG); Positive drug (melatonin) group (MTG); Tyr-Gly-Asn-Pro-Trp-Glu-Lys (TG7).

Gene Name	Gene Description	MTG vs. ICG		TG7 vs. ICG	
		log2 FC	padj	log2 FC	padj
per2	period circadian clock 2	+1.88	3.23 × 10^−9^	+2.10	6.51 × 10^−15^
gnat2	guanine nucleotide-binding protein alpha transducing activity polypeptide 2	+2.25	5.34 × 10^−9^		
per1b	period circadian clock 1b	−2.03	7.90 × 10^−9^	−2.54	2.25 × 10^−18^
cry1bb	cryptochrome circadian clock 1bb	+1.36	8.31 × 10^−9^	+1.48	1.99 × 10^−12^
dmtn	dematin actin-binding protein	−2.45	2.49 × 10^−8^		
h3f3d	H3 histone%2C family 3D	−1.11	4.98 × 10^−8^		
cry1ba	cryptochrome circadian clock 1ba	+1.58	5.83 × 10^−8^	+1.54	3.36 × 10^−9^
Si: dkey-33c14.3	Si: dkey-33c14.3	−2.22	8.09 × 10^−8^		
arr3b	arrestin 3b	+2.19	1.23 × 10^−6^	+2.70	2.95 × 10^−9^
opn1mw1	opsin 1 medium-wave-sensitive 1	+1.98	2.03 × 10^−6^	+2.54	1.98 × 10^−15^
crygmx	crystallin%2C gamma MX	−1.40	3.00 × 10^−6^		
opn1sw1	opsin 1 short-wave-sensitive 1	+1.47	4.22 × 10^−6^		
grk7a	G protein-coupled receptor kinase 7a	+2.01	5.56 × 10^−6^	+2.47	1.22 × 10^−10^
gnb3b	guanine nucleotide-binding protein beta polypeptide 3b	+1.69	5.56 × 10^−6^	+2.32	2.81 × 10^−15^
mipb	The major intrinsic protein of lens fiber b	−1.37	2.91 × 10^−5^		
fkbp5	FK506 binding protein 5			−2.31	2.00 × 10^−28^
gngt2b	guanine nucleotide-binding protein gamma transducing activity polypeptide 2b			+1.64	2.28 × 10^−13^
gabra6a	gamma-aminobutyric acid A receptor%2C alpha 6a			+2.22	2.15 × 10^−9^
hnrnpa3	heterogeneous nuclear ribonucleoprotein A3			+1.77	4.22 × 10^−9^
nadk2	NAD kinase 2			−3.88	3.67 × 10^−9^
pde6c	phosphodiesterase 6C			+2.07	3.67 × 10^−9^
sik1	salt-inducible kinase 1			−1.62	1.52 × 10^−7^

Note: −: gene down-regulation; +: gene up-regulation; log2 FC: log2 Fold Change.

## Data Availability

Data is contained within the article or Appendix A.

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
