# Peer review of "Small Peptides Isolated from Enzymatic Hydrolyzate of Pneumatophorus japonicus Bone Promote Sleep by Regulating Circadian Rhythms"

_foods, 2023, doi:10.3390/foods12030464_

Round 1

Reviewer 1 Report

Report by Wang and coworkers is an interesting finding highlighting the pharmacological role of TG7 peptide towards improvement sleep in zebrafish in vivo.

1.    Add abbreviations and full terms in figure legends e.g., blank control group (BCG), normal growth of zebrafish without adding any drugs; (2) insomnia control group (ICG), positive drug (melatonin) group (MTG) etc.

2.    The above comment is applicable to all figures

3.    Figure 6: Please increase font size and clarity of all sub figures

4.    Figure 7: Change cell nuclear membrane to nuclear membrane

Author Response

Response to Reviewer 1 Comments

To reviewer 1

1.Response to comment 1: Add abbreviations and full terms in figure legends e.g., blank control group (BCG), normal growth of zebrafish without adding any drugs; (2) insomnia control group (ICG), positive drug (melatonin) group (MTG) etc.

Response: Sincerely thanks for your advice, we have added abbreviations and full terms to the legend in the full text, the portion we added has been marked in red.

  1. Response to comment 2: The above comment is applicable to all figures.

Response: Sincerely thanks for your advice, we have added abbreviations and full terms to the legends of all figures in the article, the portion we added has been marked in red.

  1. Response to comment 3: Figure 6: Please increase font size and clarity of all sub figures.

Response: Sincerely thanks for your advice, we have increased the font size and clarity of Figure 6. Corrected as suggestion.

  1. Response to comment 4: Figure 7: Change cell nuclear membrane to nuclear membrane.

Response: Sincerely thanks for your advice, corrected as suggestion.

Reviewer 2 Report

This is an interesting study that investigated sleep promotion by small peptides isolated from enzymatic hydrolyzate of Pneumatophorus japonicus bone. Study is erll conducted, and I have only a few minor comments:

- more detailed information regarding the reason why zebrafish could be "ideal source of sleep-promoting peptides" should be provided in introduction

- limitations of this study should be better emphasized in the Discussion, as well as potential clinical implications that can be derived from the results from this study

Author Response

Response to Reviewer 2 Comments

To reviewer 2

  1. Response to comment 1: More detailed information regarding the reason why zebrafish could be "ideal source of sleep-promoting peptides" should be provided in introduction.

Response: Sincerely thanks for the suggestions on our articles, the comments you made were very constructive. In line 74-82 of this article, we have added more comprehensive information in the introduction to explain the reason why zebrafish could be "ideal source of sleep-promoting peptides". The portion we added has been marked in red.

  1. Response to comment 2: Limitations of this study should be better emphasized in the Discussion, as well as potential clinical implications that can be derived from the results from this study.

Response: Sincerely thanks for the suggestions on our articles, the comments you made were very constructive. In line 519-532 of this article, we add the limitations of this study in the Discussion and the potential clinical implications that can be drawn from the results of this study. The portion we added has been marked in red.

Reviewer 3 Report

Comments to the Author 

Manuscript No 2143330, entitled “Small peptides isolated from enzymatic hydrolyzate of Pneumatophorus japonicus bone promote sleep by regulating circadian rhythms”

General Comments:

The manuscript is interesting and but discussion is poor regarding previous work especially of polypeptide use as sleep promoting agent. 

Introduction is written nicely but write down novelty of manuscript in scientific manner on the basis of previous work done in the introduction. Novelty can be concerned to the consumers. This should also be focused.

Materials and methods are briefed in the scientific way however, some specific points are mentioned below

Specific comments:

L101-4: please mention concentration of peptide solutions, and how these concentrations were decided? 

L117: Please check enzymolysis temperature 

L118-9: please mention specifications of centrifugation and drcolorization.  

Results are elaborated nicely along with Tables and figures but discussion needs further improvement. 

Conclusion depicts findings of the study.

Author Response

Response to Reviewer 3 Comments

To reviewer 3

General Comments:

1.Response to comment 1: The manuscript is interesting and but discussion is poor regarding previous work especially of polypeptide use as sleep promoting agent. 

Response: Sincerely thanks for the suggestions on our articles, we discussed in more depth previous work and polypeptide use as sleep promoting agent.

2.Response to comment 2: Introduction is written nicely but write down novelty of manuscript in scientific manner on the basis of previous work done in the introduction. Novelty can be concerned to the consumers. This should also be focused.

Response: Sincerely thanks for the suggestions on our articles, the comments you made were very constructive. In line 65-68 of this article, we further added the novelty of this study in the Introduction. The portion we added has been marked in red.

  1. Response to comment 3: Materials and methods are briefed in the scientific way however, some specific points are mentioned below

Specific comments:

L101-4: please mention concentration of peptide solutions, and how these concentrations were decided?

Response: Sincerely thanks for your advice, we have added the concentration of the peptide solution in section 2.3. We determined the concentrations of the peptide components in pre-experiments, which were safe and sleep-promoting for zebrafish.

L117: Please check enzymolysis temperature.

Response: Sincerely thanks for your advice, corrected as suggestion.

L118-9: Please mention specifications of centrifugation and drcolorization.

Response: Sincerely thanks for your advice, we have indicated the specifications of centrifugation and drcolorization in section 2.5. The portion we added has been marked in red.

  1. Response to comment 4: Results are elaborated nicely along with Tables and figures but discussion needs further improvement.

回应:衷心感谢您的建议,我们对讨论进行了更全面的更改和补充。我们修改和添加的部分已标记为红色。

  1. 对评论5的答复:结论描述了研究结果。

回应:衷心感谢您的意见和建议。结论已经过调整和修改,以便进行提炼和总结,使其不再是结果的重复。